# Robust Stabilization of Linear Switched Systems with Unstable Subsystems

**Martín-Antonio Rodríguez-Licea** [1,†,*] ，**Francisco-J. Perez-Pinal** [2,†] **and Juan Prado Olivares** [2]

1   Departamento de Ingeniería Electrónica, CONACYT-Instituto Tecnológico de Celaya,
    Guanajuato 38010, Mexico
2   Departamento de Ingeniería Electrónica, Instituto Tecnológico de Celaya, Guanajuato 38010, Mexico;
    francisco.perez@itcelaya.edu.mx (F.-J.P.-P.); juan.prado@itcelaya.edu.mx (J.P.O.)
*   Correspondence: martin.rodriguez@itcelaya.edu.mx; Tel.: +52-014-616-117575 (ext. 5164)
†   These authors contributed equally to this work.



**Featured Application: The current analysis can be applied to a wide variety of switched systems with unstable subsystems, for instance power electronic, mechanical, aeronautic, and nuclear plant systems, among others.**

**Abstract:** This paper deals with the robust stability of a class of uncertain switched systems with possibly unstable linear subsystems. In particular, conditions for global uniform exponential stability are presented. In addition, a procedure to design a mode dependent average dwell time switching signal that stabilizes a switched linear system composed of diagonalizable subsystems is established, even if all of them are stable/unstable and time-varying (within design bounds). An illustrative example of the stabilizing switching law design and numerical results are presented.

**Keywords:** switched systems analysis; mathematically-unstable subsystems; robust control demonstration

## 1. Introduction

In recent years, the interest in Switched Linear Systems (SLS) has increased because of its capability to represent complex nonlinear systems in a more tractable math form, and their analysis has spread out as a new branch of stability and control especially for SLS with one or more unstable subsystems, while some of the Lyapunov and other theories can be applied to those with all stable subsystems [1,2].

State-dependent and time-dependent switching signals are the main approaches to design stabilizing switching laws. In the former, the whole state space is usually divided to facilitate the search for Lyapunov-like functions; unfortunately, the system's states must be measurable or observable. In the latter, time-constrained switching is used, wherein a stable subsystem is activated for enough time to stabilize the entire system [3–6].

The stability of SLS with one or more unstable subsystems is now well established from the stabilizing switching signal point of view [7–13]. However, studies on time-dependent switching stabilization with Mode-Dependent Average Dwell Time (MDADT) dedicated to the robust stability of time-varying SLS have remained unfinished.

The authors in [14] aimed at the robust stability of a discrete, positive, switched system, with bounded control inputs and with stable and unstable subsystems; the examples presented showed that a stabilizing MDADT signal can be easily designed. In [15] was presented a switching stabilization of SLS composed of both stable and unstable subsystems, easily extendable to all stable or unstable subsystems. Although the study had typographic errors, the authors presented important results on stability and the switching MDADT signal design. In [16], a parameter-dependent MDADT

switching scheme related to a set of parameter-dependent Lyapunov functions was proposed in order to control a class of switching LPVsystems; the proposed approach was applied to satisfy the overall control objectives related to a variable-sweep, wing-morphing aircraft. A full-envelope flight controller using switched linear modeling based on MDADT, with Locally-Overlapped Subsystems (LOSS), was proposed for a full-envelope flight in [17]. A stability proof by using a common Lyapunov function for each LOSS was also reported. It is worth mentioning that each active subsystem can only switch to another adjacent LOSS subsystem, and this particularity relaxed the controller, stability, and control problem formulation. In [18], multiple co-positive Lyapunov functions and an MDADT technique were combined to derive sufficient conditions for the input-output finite time stability problem. Additionally, a controller was derived, and numerical examples were also provided to show the feasibility of the proposed technique. A quasi-time dependent H-∞ robust controller for a switched system with MDADT was proposed in [19]; a Lyapunov function was used to prove theoretical stability. Numerical results in an SLS and practical results in a power electronic, boost converter were reported. Unfortunately, the switching signal to control both systems had a variable frequency, which in the case of the boost converter produced noise and EMI degradation. The stability analysis of time-varying impulsive positive hybrid systems with time-varying, distributed delays, and all unstable subsystems, by using an MDADT, was reported in [20]. Additionally, a concept of input-output finite time stability was presented. Finally, their numerical simulations showed the feasibility of the proposed method. It is worth mentioning the important contribution in the topic of adaptation and robustness of SLS in [21]. This work was focused on four main areas: adaptive tracking using extended and average dwell times, adaptive asymptotic tracking, robust adaptive tracking, adaptive stabilization with time-varying delays, and robust stability and stabilization with switching delays.

It can be noticed from the above state of the art that the robust stability and stabilizing switching MDADT signal design for SLS with any combination of stable/unstable systems is a topic of recent interest for researchers and industry. Therefore, in this paper, the robust stability for diagonalizable uncertain SLS is analyzed, and a new result is presented. Such analysis allows the design of a stabilizing switching MDADT signal, and it is not restricted to positive systems, nor to all stable/unstable subsystems.

Although this paper is inspired by [15], a generalization to uncertain SLSs is demonstrated by analyzing the uncertain polytope for the design of a stabilizing switching law, and an additional simplification of the conditions for stability is obtained. An illustrative example is presented, and a numerical analysis complements the proposed approach.

This paper is organized as follows. Section 2 is aimed at the math preliminaries. The principal result is presented in Section 3 and the numerical example in Section 4. Finally, conclusions are presented in the last section.

## 2. Preliminaries

In this paper, $\mathbb{R}$ represents the real numbers' set, $\mathbb{I}$ represents the positive integers' set excluding zero, $\mathbb{I}_n$ represents the set of integers $\{ 1, ..., n \}$ where $n$ is the dimension of the system, $Q$ is the identity matrix of adequate dimensions, $\mathbb{Z}$ represents the positive integers' set including zero, and the set operations are denoted as follows: $\subset, \subseteq$ for subset and strict subset, respectively, $\cup, \cap$ for union and intersection, respectively, $\oplus$ for addition, and $\mathcal{C}$ for the complement. $Re(\cdot), Max(\cdot), Min(\cdot)$ stands for the real part, the maximum of real parts, and the minimum of real parts of a real or complex number. $\lambda(\cdot)$ and $\delta(\cdot)$ are the eigenvalues and singular values of a matrix. $\| \cdot \|$ and $\| \cdot \|_r$ denote the spectral norm and the spectral norm with restriction to a subspace $r$, respectively.

Consider a Linear Time-Variant Switched System (LTVSS) as follows:

$$\dot{x}(t) = A_{\sigma}(t)x(t) \tag{1}$$

where $x(t) \in \mathbb{R}^n$ is the vector state, $A_\sigma(t) \in \mathbb{R}^{n \times n}$ is a time-varying matrix that can commute (e.g., from $A_1(t_1)$ to $A_3(t_2)$ to $A_2(t_3)$...), and $\sigma(t) : t \to I$ is the stabilization switching signal to be designed ($I = \{1, 2, ..., s\} \subset \mathbb{I}$, where $s$ is the number of subsystems). For a switching sequence $t_1, t_2, ...$, $\sigma(t)$ is a piece-wise continuous function and $\sigma(t) = p \in I$, $\forall t \in [t_i, t_{i+1})$ with $i \in I$ and $t_i < t_{i+1} < t_{i+2}, ...$

The total number of time-varying entries of $A_p(t)$ is denoted as $v$; if all of the entries of $A_p(t)$ are time varying, $v = 2^{n \times n}$. With $k \leq n$ denoting the row and $l \leq n$ the column of a matrix, an entry of $A_p(t)$ is denoted as $a_{[p,k,l]}(t)$, and if it is known that $\underline{a}_{[p,k,l]} \leq a_{[p,k,l]}(t) \leq \overline{a}_{[p,k,l]}$, each $p^{\text{th}}$ subsystem can be written as a Linear Parameter-Variant Subsystem (LPVS) by a polytopic simplice representation [22]:

$$\dot{x}(t) = A_p(t)x(t) = A_p(\theta(t))x(t) = \left( \sum_{j=1}^{2^v} \theta_j(t) A_{[p,j]} \right) x(t) \tag{2}$$

where $\theta \in \Theta = \left\{ \theta \,|\, 0 \leq \theta_j(t) \leq 1, \sum \theta_j = 1, \forall j \in V = \{1, ..., 2^v\} \right\}$, and $A_{[p,j]}$ is the $j^{\text{th}}$ vertex of $A_p(\theta(t))$. For instance, the vertexes can be built with the combinations of $\underline{a}_{[p,k,l]}$ and $\overline{a}_{[p,k,l]}$:

$$A_{p,1} = \begin{bmatrix} \underline{a}_{[p,1,1]} & \underline{a}_{[p,1,2]} & \cdots \\ \underline{a}_{[p,2,1]} & \underline{a}_{[p,2,2]} & \cdots \\ \vdots & \vdots & \ddots \end{bmatrix}, A_{p,2} = \begin{bmatrix} \overline{a}_{[p,1,1]} & \underline{a}_{[p,1,2]} & \cdots \\ \underline{a}_{[p,2,1]} & \underline{a}_{[p,2,2]} & \cdots \\ \vdots & \vdots & \ddots \end{bmatrix}, A_{p,3} = \begin{bmatrix} \underline{a}_{[p,1,1]} & \overline{a}_{[p,1,2]} & \cdots \\ \underline{a}_{[p,2,1]} & \underline{a}_{[p,2,2]} & \cdots \\ \vdots & \vdots & \ddots \end{bmatrix}, \cdots \tag{3}$$

Below are definitions and previous results to be used.

**Definition 1** ([23]). *Suppose $A \in \mathbb{R}^{n \times n}$, and $S \subseteq \mathbb{R}^n$ is a subspace. $S$ is $A$-invariant if $AS \subseteq S$, that is, $\forall b \in S \Rightarrow Ab \in S$.*

In the following, $\sigma(t)$ is considered a strict piecewise continuous (must commute), mode-dependent average dwell time function; that is, each mode has its own average dwell time in order to obtain more flexible and less conservative stability conditions, in comparison with analyzes that use a single average dwell time [24]:

**Definition 2.** *For a switching signal $\sigma(t)$ with $T \geq t \geq 0$, let $N_{[\sigma,p]}(T, t)$ be the quantity of switching events (in [24], this term was originally called switching numbers; in this paper, this is changed for clearness to the quantity of switching events) that the $p^{\text{th}}$ subsystem is activated along the interval $[t, T]$ and $\mathcal{T}_p(T, t)$ the total running time of the $p^{\text{th}}$ subsystem over the interval $[t, T]$, $p \in I$. We say that $\sigma(t)$ has a mode-dependent average dwell time $\tau_p$ if there exist $N_{[0,p]} > 0$ and $\tau_p > 0$ such that:*

$$N_{[\sigma,p]}(T, t) \leq N_{[0,p]} + \frac{\mathcal{T}_p(T, t)}{\tau_p}, \quad \forall T \geq t \geq 0 \tag{4}$$

In this paper, a mode-dependent average dwell time switching signal is denoted as $\sigma(t) \in F_{MDADT}[N_{[0,p]}, \tau_p]$.

**Definition 3.** *The equilibrium of $x = 0$ of (2) is Globally Uniformly Exponentially Stable (GUES) under a certain switching signal $\sigma(t)$ if for initial conditions $x(t_0)$, there exist constants $\eta_1 > 0$, $\eta_2 > 0$ such that the solution of the system satisfies $\|x(t)\| \leq \eta_1 e^{-\eta_2(t - t_0)} \|x(t_0)\|$, $\forall t \geq t_0$.*

## 3. Main Result

The whole state space can be divided into two subspaces $S_p^S$ and $S_p^U$ defined as follows:

**Definition 4.** *The stable subspace $S_p^S$, $p \in I$, is spanned by the eigenvectors corresponding to the eigenvalues:*

$$\lambda_k \left( \sum_{j=1}^{2^v} \theta_j(t) A_{[p,j]} \right),$$

$$k \in \mathbb{K}_p^s = \{ m \in \mathbb{I}_n | \, Re \left( \lambda_m \left( \sum_{j=1}^{2^v} \theta_j(t) A_{[p,j]} \right) \right) \geq 0, p \in I, \theta \in \Theta, T \geq t \geq 0 \}.$$

**Definition 5.** *The unstable subspace $S_p^U$, $p \in I$, is spanned by the eigenvectors corresponding to the eigenvalues:*

$$\lambda_k \left( \sum_{j=1}^{2^v} \theta_j(t) A_{[p,j]} \right),$$

$$k \in \mathbb{K}_p^u = \{ m \in \mathbb{I}_n | \, Re \left( \lambda_m \left( \sum_{j=1}^{2^v} \theta_j(t) A_{[p,j]} \right) \right) < 0, p \in I, \theta \in \Theta, T \geq t \geq 0 \}.$$

On the other hand, the following proposition will be used later to demonstrate the main theorem:

**Lemma 1.** *Consider the switched linear system (2). If S is $A_p(\theta(t))$-invariant $\forall p \in I$ and $\forall \theta \in \Theta$, then S is $e^{A_p(\theta(t))t}$-invariant $\forall p \in I$, $\forall \theta \in \Theta$, and $\forall t \geq 0$.*

**Proof.** Consider a fixed $\theta$ and $p = 1$ for a $\Delta$ period, then $A_p(\theta) = A_{[p,\Delta]} = A_{[1,1]}$. S is $A_{[1,1]}$-invariant $\forall x \in S$ by assumption; for any sequence of events in $[p, \Delta] \in \mathbb{Z}^2$, $p \in I$, $\Delta \in \mathbb{Z}$:

$$A_{[p,\Delta]}^r x = A_{[p,\Delta]}^{r-1} A_{[p,\Delta]} x \tag{5}$$

with $x_1 = A_{[p,\Delta]} x \in S$,

$$A_{[p,\Delta]}^r x = A_{[p,\Delta]}^{r-1} x_1 \tag{6}$$

with $x_2 = A_{[p,\Delta]} x_1 \in S$,

$$A_{[p,\Delta]}^r x = A_{[p,\Delta]}^{r-2} x_2 \tag{7}$$

$$\vdots \tag{8}$$

with $x_\Delta = A_{[p,\Delta]} x_{r-1} \in S$,

$$A_{[p,\Delta]}^r x = A_{[p,\Delta]} x_r \in S \tag{9}$$

From the matrix exponential definition:

$$e^{A_{[p,\Delta]}\Delta} x = Qx + \Delta A_{[p,\Delta]} x + \frac{\Delta^2}{2!} A_{[p,\Delta]}^2 x + ... + \frac{\Delta^r}{r!} A_{[p,\Delta]}^2(\theta) x + ... \tag{10}$$

For sufficiently small values of $\Delta$ in a succession $[p_1, \Delta_1], [p_2, \Delta_2], [p_3, \Delta_3], ...$ (without loss of generality), one has the decomposition:

$$e^{A_p(\theta(t))t} x = \left( Q + \Delta_1 A_{[p_1,\Delta_1]} + \Delta_2 A_{[p_2,\Delta_2]} + ... + \frac{\Delta_1^2 A_{[p_1,\Delta_1]}^2}{2!} + \frac{\Delta_2^2 A_{[p_2,\Delta_2]}^2}{2!} + ... + \frac{\Delta_1^3 A_{[p_1,\Delta_1]}^3}{3!} + ... + \right.$$
$$\left. \Delta_1 \Delta_2 A_{[p_1,\Delta_1]} A_{[p_2,\Delta_2]} + \Delta_1 \Delta_3 A_{[p_1,\Delta_1]} A_{[p_3,\Delta_3]} + ... + \Delta_1 \Delta_2^2 A_{[p_2,\Delta_2]}^2 A_{[p_2,\Delta_2]} + ... \right) x \in S \tag{11}$$

where properties for the sum and intersection of subsets are used to complete the proof. □

This last property is known and demonstrated, in a different way, by some authors as the cocycle property (see for instance [25]).

**Lemma 2.** *Consider the subsystem* $\dot{x} = A_p(\theta(t))x$ *and let:*

$$\lambda_p^m = \left\{ Max \left( \lambda_k(A_{[p,j)]}) \right) \middle| Re \left( \lambda_k \left( A_{[p,j)]} \right) \right) < 0, p \in I, j \in \{1, ..., 2^v\}, k \in \mathbb{I}_n \right\}$$

*and:*

$$\lambda_p^M = \left\{ Max \left( \lambda_k(A_{[p,j)]}) \right) \middle| Re \left( \lambda_k \left( A_{[p,j)]} \right) \right) \geq 0, p \in I, j \in \{1, ..., 2^v\}, k \in \mathbb{I}_n \right\}$$

*then there exists a constant* $\epsilon_p > 0$ *such that:*

$$\left\| e^{A_p(\theta(t))t} \right\|_{S^S} \leq e^{\epsilon_p + \lambda^m t} \tag{12}$$

$$\left\| e^{A_p(\theta(t))t} \right\|_{S^U} \leq e^{\epsilon_p + \lambda^M t} \tag{13}$$

*where* $S^S$ *and* $S^U$ *are the stable and unstable subspaces of* $A_p(\theta(t))$*, respectively.*

**Proof.** Choosing from the set of diagonalizing matrices $T_{[S,\ell]}$, composed of the basis of $S^S$ (a matrix for each stable vertex in the stable subspace), one has:

$$\left\| e^{A(\theta(t))t} \right\|_S \leq T_{[S,M1]} T_{[S,M2]} \left\| Diag \left( e^{A_{[p_s,j]}} \right) \right\| \leq T_{[S,M1]} T_{[S,M2]} e^{\lambda^m} \tag{14}$$

where $T_{[S,M1]} = max \left\{ \left\| T_{[S,\ell]} \right\| \right\}$, and $T_{[S,M2]} = max \left\{ \left\| T_{[S,\ell]}^{-1} \right\| \right\}$.

On the other hand, choosing from the set of diagonalizing matrices $T_{[U,\ell]}$, composed of the basis of $S^U$ (a matrix for each unstable vertex in the unstable subspace), the worst vertex one has:

$$\left\| e^{A(\theta(t))t} \right\|_U \leq T_{[U,M1]} T_{[U,M2]} \left\| Diag \left( e^{A_{[p_u,j]}} \right) \right\| \leq T_{[U,M1]} T_{[U,M2]} e^{\lambda^M} \tag{15}$$

where $T_{[U,M1]} = max \left\{ \left\| T_{[U,\ell]} \right\| \right\}$, and $T_{[U,M2]} = max \left\{ \left\| T_{[U,\ell]}^{-1} \right\| \right\}$.

Setting:

$$\epsilon_p = Max \left\{ Ln \left( T_{[S,M1]} T_{[S,M2]} \right), Ln \left( T_{[U,M1]} T_{[U,M2]} \right) \right\} \tag{16}$$

completes the proof. □

**Theorem 1.** *Consider the switched linear polytopic system* (2). *For given constants:*

$$0 < \lambda_p^M + \frac{\epsilon_p}{\tau_p} < \alpha_p, \tag{17}$$

$$\lambda_p^m + \frac{\epsilon_p}{\tau_p} < \beta_p < 0, \tag{18}$$

$p \in I$, *and* $\theta \in \Theta$. *If there exist two sets* $I_1, I_2 \subset I$ *with* $I_1 \cup I_2 = I$ *such that* $\Omega_1 = \sum_{p \in I_1} S_p^U$ *and* $\Omega_2 = \cap_{p \in I_2} S_p^S$ *are* $A_p(\theta(t))$*-invariant* $\forall p \in I$, $\mathcal{C}(\Omega_1) \supseteq \cap_{p \in I_2} S_p^S$, *then the system* (2) *is GUES for any switching signal* $\sigma(t) \in F_{MDADT}[N_{0p}, \tau_p]$ *satisfying:*

$$\sum_{p \in I_1} \alpha_{p \in I_1} \mathcal{T}_p(T, 0) < - \sum_{p \in I_2} \beta_p \mathcal{T}_p(T, 0) - \sum_{p \in I} N_{[0,p]} \epsilon_p. \tag{19}$$

**Proof.** It is obvious that $\Omega_1$ and $\Omega_2$ are two subspaces in $\mathbb{R}^n$, and it is also clear from the definitions of $\Omega_1$ and $\Omega_2$ that:

$$\Omega_1 \cap \Omega_2 = \varnothing, \tag{20}$$

$$\mathcal{C}(\Omega_2) = \mathcal{C}(\cap_{p \in I_1} S_p^S) = \sum_{p \in I_1} \mathcal{C}(S_p^S) = \sum_{p \in I_1} S_p^U = \Omega_1 \tag{21}$$

which implies:

$$\Omega_1 \oplus \Omega_2 = \mathbb{R}^n \tag{22}$$

For any sufficiently large $T > 0$, let $t_0 = 0$ and $t_1, t_2 ..., t_i, t_{i+1}, ... t_{N_T}$ denote the switching times on the interval $[0, T]$. With initial condition $x(0) \in \mathbb{R}^n$, Lemma 1 yields:

$$x(T) = e^{(T - t_{N_T}) A_{[\theta(t), \sigma(t_{N_T})]}} ... e^{(t_{i+1} - t_i) A_{[\theta(t), \sigma(t_i)]}} ... e^{(t_1 - t_0) A_{[\theta(t), \sigma(t_0)]}} x(0) \tag{23}$$

Therefore, using (4) and Lemma 2:

$$
\begin{aligned}
\|x(T)\| \;\leq\; & \prod_{q \in \Phi_1} \left\| e^{(t_{q+1} - t_q) A_{[\sigma(t_q), j]}} \right\|_{\Omega_1} \prod_{q \in \Phi_1} \left\| e^{(t_{q+1} - t_q) A_{[\sigma(t_q), j]}} \right\|_{\Omega_2} \|x(0)\| \\
\leq\; & \prod_{q \in \Phi_1} \left\| e^{(t_{q+1} - t_q) A_{[\sigma(t_q), j]}} \right\|_{S_p^U} \prod_{q \in \Phi_1} \left\| e^{(t_{q+1} - t_q) A_{[\sigma(t_q), j]}} \right\|_{S_p^S} \|x(0)\| \\
\leq\; & \prod_{p \in I_1} e^{N_{[\sigma,p]}(T,0)\epsilon_p} e^{\lambda_p^M \mathcal{T}_p(T,0)} \prod_{p \in I_2} e^{N_{[\sigma,p]}(T,0)\epsilon_p} e^{\lambda_p^m \mathcal{T}_p(T,0)} \|x(0)\| \\
=\; & e^{\sum_{p \in I} N_{[\sigma,p]}(T,0)\epsilon_p} e^{\sum_{p \in I_1} \lambda_p^M \mathcal{T}_p(T,0) + \sum_{p \in I_2} \lambda_p^m \mathcal{T}_p(T,0)} \|x(0)\| \\
\leq\; & e^{\sum_{p \in I} N_{[0,p]}\epsilon_p} e^{\sum_{p \in I_1} \lambda_p^M \mathcal{T}_p(T,0) + \sum_{p \in I_2} \lambda_p^m \mathcal{T}_p(T,0) + \sum_{p \in I} \frac{\epsilon_p \mathcal{T}_p(T,0)}{\tau_p}} \|x(0)\| \\
\leq\; & e^{\sum_{p \in I} N_{[0,p]}\epsilon_p} e^{\sum_{p \in I_1} \left(\lambda_p^M + \frac{\epsilon_p}{\tau_p}\right)\mathcal{T}_p(T,0) + \sum_{p \in I_2} \left(\lambda_p^m + \frac{\epsilon_p}{\tau_p}\right)\mathcal{T}_p(T,0)} \|x(0)\| \tag{24}
\end{aligned}
$$

where $\Phi_1, \Phi_2$ denote the sets of $q$ satisfying $\sigma(t_q) \in I_1, I_2$, respectively. Therefore, if:

$$\tau_p \geq \frac{\epsilon_p}{\alpha_p - \lambda_p^M}, \forall p \in I_1 \tag{25}$$

$$\tau_p \geq \frac{\epsilon_p}{\beta_p - \lambda_p^m}, \forall p \in I_2 \tag{26}$$

is used, then:

$$\|x(T)\| \;\leq\; e^{\sum_{p \in I} N_{[0,p]}\epsilon_p} e^{\sum_{p \in I_1} \alpha_p \mathcal{T}_p(T,0) + \sum_{p \in I_2} \beta_p \mathcal{T}_p(T,0)} \|x(0)\|$$

which means that the system is GUES under MDADT, satisfying (19). □

It is worth mentioning that this theorem can be used even if all of the subsystems are stable; in such a case, $I_1 = \varnothing$, $I_2 = I$, $\Omega_1 = \varnothing$, $\Omega_2 = \mathbb{R}^n$. Even more, the result can be used even if all of the subsystems are unstable; in such a case, $I_2 = \varnothing$, $I_1 = I$, $\Omega_2 = \varnothing$, $\Omega_1 = \mathbb{R}^n$.

## 4. Simulations

In this section is illustrated the main result of this paper through MATLAB simulations. For simplicity and demonstrative reasons, the following system is proposed:

$$\dot{x}(t) = A_{\sigma}(t)x(t) = A_{\sigma}(\theta(t))x(t) = \left( \sum_{j=1}^{4} \theta_j(t) A_{[\sigma,j]} \right) x(t), \sigma(t) : \mathbb{R} \rightarrow \{1, 2\} \tag{27}$$

with vertexes:

$$A_{1,1} = \begin{bmatrix} 0.1074 & 0.0154 \\ -0.0154 & 0.2026 \end{bmatrix}, A_{1,2} = \begin{bmatrix} 0.0966 & 0.0206 \\ -0.0206 & 0.2234 \end{bmatrix}, A_{1,3} = \begin{bmatrix} 0.1069 & 0.0189 \\ -0.0189 & 0.2231 \end{bmatrix},$$

$$A_{1,4} = \begin{bmatrix} 0.0971 & 0.0171 \\ -0.0171 & 0.2029 \end{bmatrix}, A_{2,1} = \begin{bmatrix} -4.9701 & -0.1995 \\ 0.1995 & -6.3299 \end{bmatrix}, A_{2,2} = \begin{bmatrix} -5.4816 & -0.1228 \\ 0.1228 & -6.3184 \end{bmatrix},$$

$$A_{2,3} = \begin{bmatrix} -4.9556 & -0.2962 \\ 0.2962 & -6.9744 \end{bmatrix}, A_{2,4} = \begin{bmatrix} -5.4671 & -0.2194 \\ 0.2194 & -6.9629 \end{bmatrix},$$

In Figure 1 is shown the dynamic behavior for the system (27), with nominal parameter values and the switching law plotted in Figure 2. Note that while the first state converges to zero, the second state is not GUES, and a switching law is designed based on the main result of this paper; the objective is to design a switching signal $p = \sigma(t) \in F_{MDADT}[N_{[0,p]}, \tau_p]$ such that (27) is GUES.

From Lemma 2, $\lambda_1^M = 0.22$, $\lambda_1^m = \emptyset$, $\lambda_2^M = \emptyset$, $\lambda_2^m = -5.00$,

$$T_{[U,1]} = \begin{bmatrix} 0.90 & 0.15 \\ 0.15 & 0.90 \end{bmatrix}, \tag{28}$$

$$T_{[U,2]} = \begin{bmatrix} 1 & 0.15 \\ 0.15 & 1 \end{bmatrix}, \tag{29}$$

$\epsilon_p = 0.6931 \ \forall p$. Set $\mathcal{T}_p(T,t) = 26s$, $N_{[\sigma,1]}(T,t) = N_{[\sigma,2]}(T,t) = 26$. For (4) and using $\tau_1, \tau_2 \le 1$, then $N_{[0,1]} = N_{[0,2]} = 1$. From (17)–(19):

$$-5 + \frac{0.6931}{\tau_2} < \beta_2 < 0 \tag{30}$$

$$0 < 0.22 + \frac{0.6931}{\tau_1} < \alpha_1 \tag{31}$$

$$26\alpha_1 < -26\beta_2 - 1.3862 \tag{32}$$

With $\tau_1 = \tau_2 = 0.5s$, $\beta_2 = 3.5$ and $\alpha_1 = 1$ are selected, and all the conditions of Theorem 1 are satisfied.

Under the above designed switching law, simulations that include the introduction of perturbations in the entries of $A_p(\theta)$ in aleatory sequences are performed. The entries are stepped between their maximum and minimum values in order to illustrate the validity of the analysis.

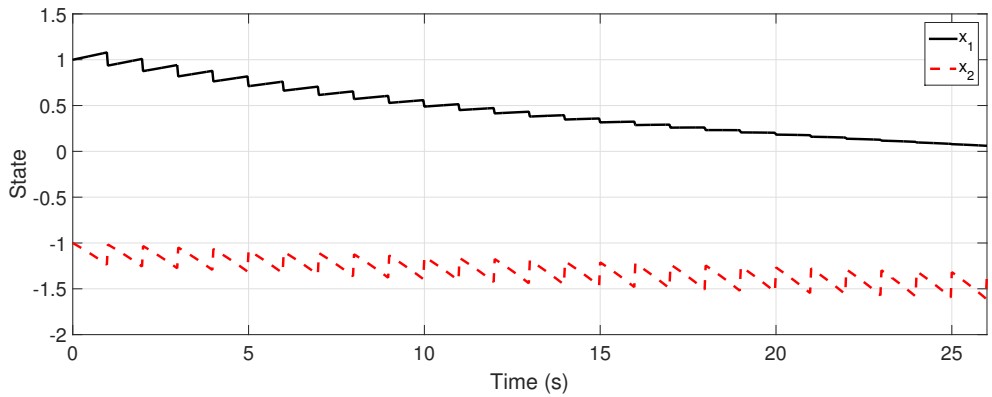

**Figure 1.** System trajectories for an arbitrary switching law.

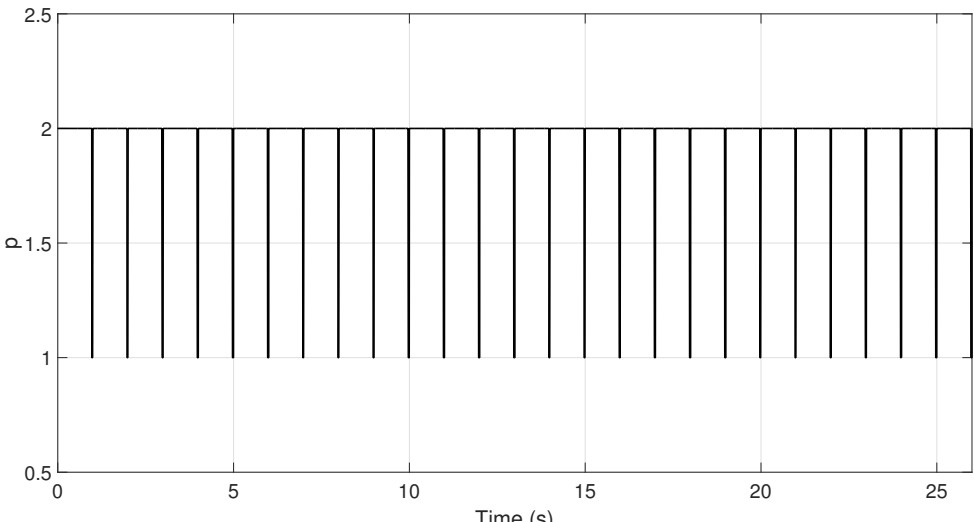

**Figure 2.** Arbitrary switching law that unstabilizes the system's trajectories.

In Figure 3 are shown the system's trajectories for an initial condition $x_1(0) = 1$ and $x_2(0) = -1$, with the designed switching law of Figure 4 and with the $A_p(\theta)$ entries' changes plotted in Figure 5. Note that even under hard parameter changes (abrupt changes), the system's trajectories converge to zero.

Finally, the switching law is designed for the above example, with the approach in [15] in order to show comparative results. Such an approach is selected to provide the fairest comparison: other approaches, do not provide a switching signal design method and/or involve discrete systems, state estimation, fuzzy logic, LMIresolution, etc. Recall that the approach in [15] is not intended to be robust against abrupt parameter changes.

Using the nominal parameters according to [15], it is possible to find a diagonalizing matrix such that $\epsilon_1 = 0.3365$, $\epsilon_2 = 0.3023$, $\lambda_1^M = 0.2$, and $\lambda_2^m = 5.25$. Selecting $\tau_1 = 0.5s$, it is obtained that $\tau_2 \geq 2.42\ s$ such that $\tau_2 = 5s$, and all of the stability conditions are met for the nominal system. In Figure 6 is shown the state behavior for the same initial conditions of the previous example ($x_1(0) = 1$ and $x_2(0) = -1$) and with the $A_p(\theta)$ entries' changes plotted in Figure 5; since the time restrictions for dwell times are more lax in such an approach, the transient state is longer than that obtained with the approach of this paper (Figure 3). In Figure 7 are shown the phase portraits for both approaches comparatively; note that the convergence with the presented approach in this paper is achieved smoothly.

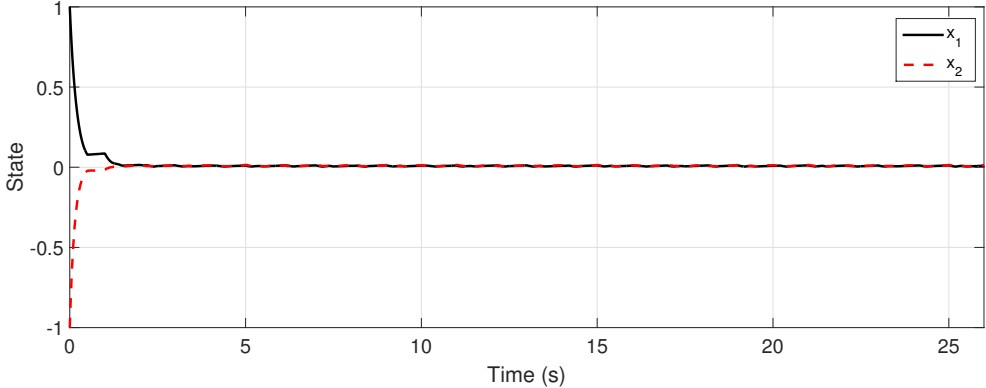

**Figure 3.** System trajectories, under arbitrary changes in parameters/entries of the system's matrix and the designed switching law.

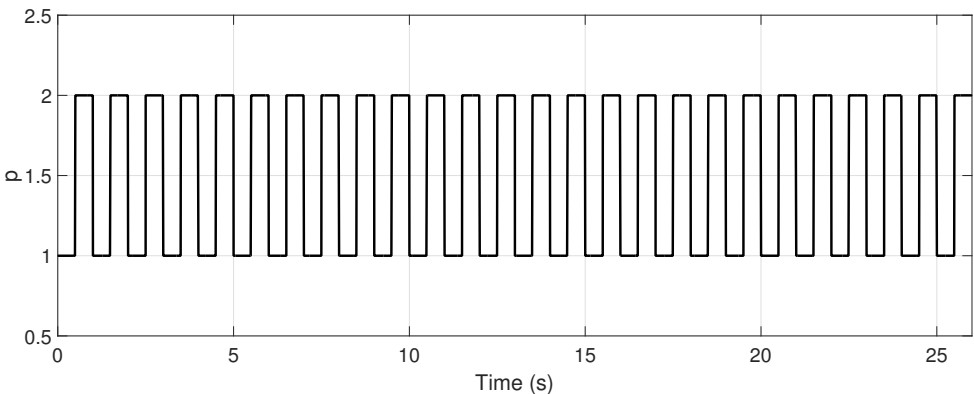

**Figure 4.** Designed switching law.

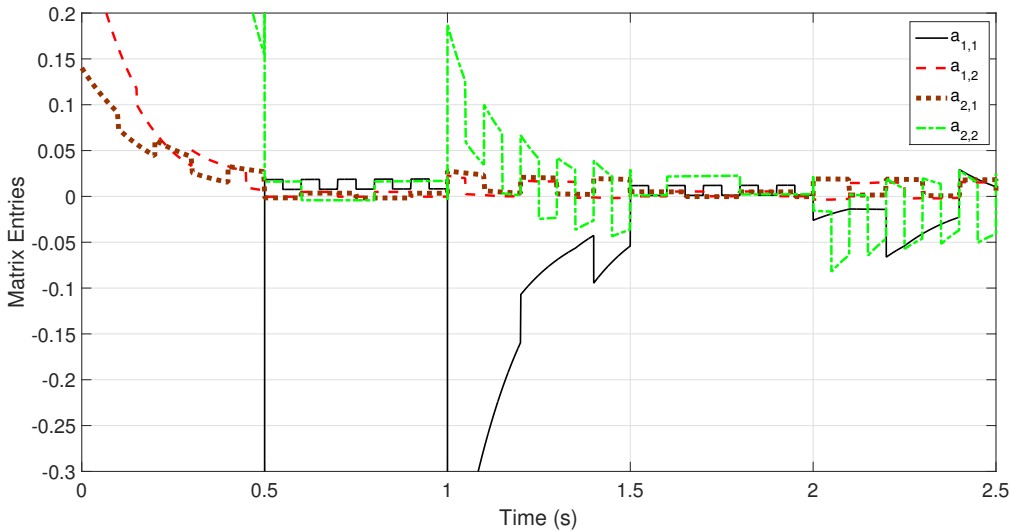

**Figure 5.** Arbitrary changes in parameters/entries of the system's matrix.

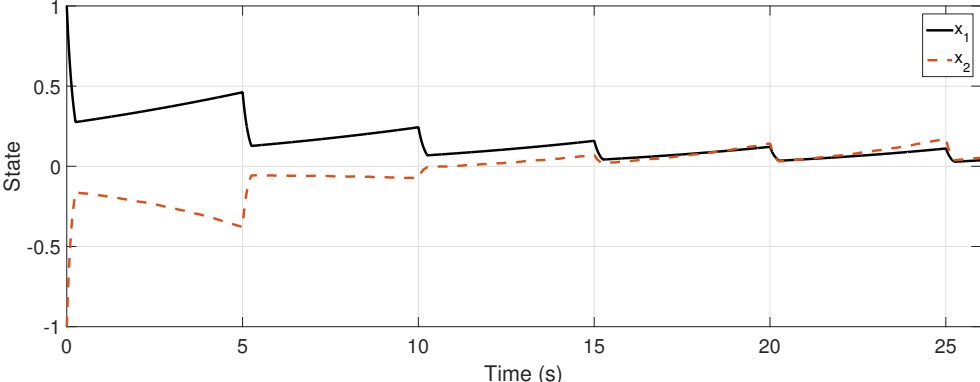

**Figure 6.** System trajectories, under arbitrary changes in parameters/entries of the system's matrix with the switching law designed in [15].

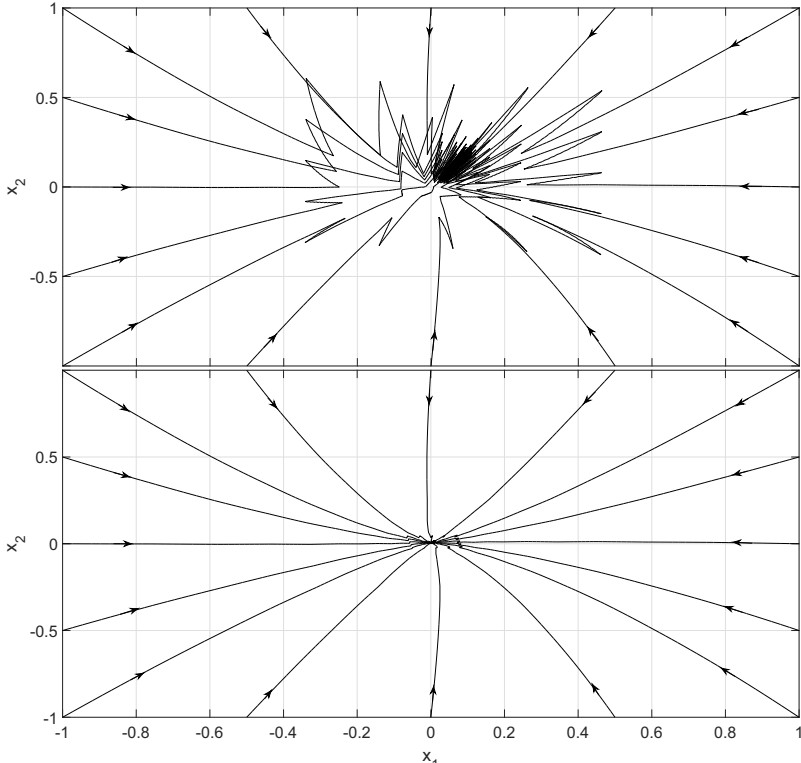

**Figure 7.** Phase portrait comparing [15] (upper plot) and the approach of this paper (lower plot).

## 5. Conclusions

This paper is aimed at the robust asymptotic stability of a class of time-variant switched linear systems composed of stable and unstable subsystems or all stable/unstable subsystems.

The main result in this paper, allows the design of a switching law ensuring the asymptotic stability; this is exemplified with numerical results that include abrupt, but bounded changes in the parameters and a comparison with similar (not robust) approaches, illustrating that the designed switching law smoothly stabilizes the parameter-varying switched system.

**Author Contributions:** Conceptualization, M.-A.R.-L.; methodology, M.-A.R.-L. and F.-J.P.-P.; software, M.-A.R.-L. and J.P.O.; validation, M.-A.R.-L. and F.-J.P.-P.; formal analysis, M.-A.R.-L.; investigation, M.-A.R.-L. and F.-J.P.-P.; writing, original draft preparation, M.-A.R.-L.; writing, review and editing, M.-A.R.-L. and J.P.O.

**Funding:** This research was funded by CONACYT México Grant Number Cátedra 4155.

**Conflicts of Interest:** The authors declare no conflict of interest.

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
