# Peer review of "Robust Stabilization of Linear Switched Systems with Unstable Subsystems"

_applsci, doi:10.3390/app8122620_

Round 1
Reviewer 1 Report
The considered class is very restrictive as compared to works in Automatica or TAC, but there is a small extension to robustness of SLS which in my opinion justifies publication
English is not good and should be extensively revised.
Author Response
Response to the suggestions and comments on Manuscript ID 377824 entitled “Robust Stabilization of Linear Switched Systems with Unstable Subsystems”
The authors thank the honorable Editor-in-Chief (EIC) and Associate Editor (AE) for giving an opportunity to incorporate the valuable suggestions given by reviewers, thereby improving the quality of the paper. The suggestions given by the reviewers are incorporated in the revised manuscript. The authors hope this revision will make our manuscript to meet the requirements of the journal. Changes due to the suggestions and comments of the reviewers are marked in yellow on the new version of the paper and in the following we answer to all the comments of the reviewers; the reviewer’s comments are numbered and indicated with bold and blue fonts as originally were sent.
The authors thank the Honorable Reviewer 1 for his excellent review of the paper and in-depth suggestions made to improve the quality of the paper. Followings are the responses to the suggestions of the Honorable Reviewer. The valuable suggestions and corrections are incorporated cautiously, and the paper is made clearer while revising the manuscript.
1.- English is not good and should be extensively revised.
Honorable Reviewer, thank you for the comment. We have made a thorough revision of the English grammar in the new version of the paper.

Reviewer 2 Report
The paper presents a new stabilising design technique for diagonalisable uncertain switching systems with possibly unstable subsystems. Although the paper is inspired by reference [15], it contains some original contributions that might deserve publication. However, the presentation, with particular regard to the English, is not always clear and, in this reviewer’s opinion, should be improved before possible publication. Few indications on how to revise the paper follow.
- Title: it is too long and badly phrased.
- Abstract: The right location of the first three sentences is the Introduction, not the abstract. The remaining sentences are poorly written. For example, the fourth sentence could be rephrased as follows: “This paper deals with the robust stability of a class of uncertain switched systems with possibly unstable linear subsystems. In particular, a condition for …..”
- Introduction: After a fairly extensive review of the literature on Switched Linear Systems (SLS), the specific contribution of the paper should be pointed out more explicitly (it is not enough to say that the paper “differs considerably from such paper” (i.e., from [15]).
- Preliminaries: Formula (1) should be introduced in a more appropriate way. The phrase “in state space representation” is superfluous at the point where it is now since x has already been named “state vector”. In Definition 2 the phrase “occasions that the p-th subsystem is activated ..” is not grammatically correct. Also the following sentences “For simplicity it is denoted …” and “For (sic!) the following … “ are not correct.
- Main result: Complete the first sentence with “defined as follows.” and introduce the subspaces in a proper way. Avoid repeating the word “result” in the same sentence. Replace “by premise” by “by assumption”? Replace “With sufficiently small ” by “For sufficiently small”. In Lemma 2: write “there exists” instead of “there exist”. In Theorem 1: write “system (2)” instead of “system 2”. In its proof cancel the comma before formula (20).
- Simulations: Rewrite the first sentence (first subject, then verb etc.) as well as the sentence after the vertices. Modify the expression “simulation is presented which consists of …”. Make the sentences smoother.
- Conclusions: again, sentence reordering is required.
Check and complete the references.
Finally, pay attention to punctuation.
Author Response
Response to the suggestions and comments on Manuscript ID 377824 entitled “Robust Stabilization of Linear Switched Systems with Unstable Subsystems”
The authors thank the honorable Editor-in-Chief (EIC) and Associate Editor (AE) for giving an opportunity to incorporate the valuable suggestions given by reviewers, thereby improving the quality of the paper. The suggestions given by the reviewers are incorporated in the revised manuscript. The authors hope this revision will make our manuscript to meet the requirements of the journal. Changes due to the suggestions and comments of the reviewers are marked in yellow on the new version of the paper and in the following we answer to all the comments of the reviewers; the reviewer’s comments are numbered and indicated with bold and blue fonts as originally were sent.
The authors thank the Honorable Reviewer 2 for his excellent review of the paper and in-depth suggestions made to improve the quality of the paper. Followings are the responses to the suggestions of the Honorable Reviewer. The valuable suggestions and corrections are incorporated cautiously, and the paper is made clearer while revising the manuscript. Thank you.
1.- However, the presentation, with particular regard to the English, is not always clear and, in this reviewer’s opinion, should be improved before possible publication.
Honorable Reviewer, thank you for the comment. We have made a thorough revision of the English grammar in the new version of the paper.
2.- Title: it is too long and badly phrased.
Honorable Reviewer, thank you for the comment. We have changed the title to “Robust Stabilization of Linear Switched Systems with Unstable Subsystems”. Except for your best opinion, we consider that the title is more appropriate.
3.- Abstract: The right location of the first three sentences is the Introduction, not the abstract. The remaining sentences are poorly written. For example, the fourth sentence could be rephrased as follows: “This paper deals with the robust stability of a class of uncertain switched systems with possibly unstable linear subsystems. In particular, a condition for …..”
Honorable Reviewer, thank you for the comments. We have taken your recommendations.
4.- Introduction: After a fairly extensive review of the literature on Switched Linear Systems (SLS), the specific contribution of the paper should be pointed out more explicitly (it is not enough to say that the paper “differs considerably from such paper” (i.e., from [15]).
Honorable Reviewer, thank you for the comment; we agree with you. This paragraph was modified in order to be more explicit.
5.- Preliminaries: Formula (1) should be introduced in a more appropriate way. The phrase “in state space representation” is superfluous at the point where it is now since x has already been named “state vector”. In Definition 2 the phrase “occasions that the p-th subsystem is activated ..” is not grammatically correct. Also the following sentences “For simplicity it is denoted …” and “For (sic!) the following … “ are not correct.
Honorable Reviewer, thank you for the comments. In the new version of the paper, these sentences were modified.
6.- Main result: Complete the first sentence with “defined as follows.” and introduce the subspaces in a proper way. Avoid repeating the word “result” in the same sentence. Replace “by premise” by “by assumption”? Replace “With sufficiently small ” by “For sufficiently small”. In Lemma 2: write “there exists” instead of “there exist”. In Theorem 1: write “system (2)” instead of “system 2”. In its proof cancel the comma before formula (20).
Honorable Reviewer, thank you for the comments. We agree with your suggestions and in the new version of the paper the mentioned phrases were corrected.
7.- Simulations: Rewrite the first sentence (first subject, then verb etc.) as well as the sentence after the vertices. Modify the expression “simulation is presented which consists of …”. Make the sentences smoother.
Honorable Reviewer, thank you for the comments. In the new version of the paper, these phrases were corrected.
8.- Conclusions: again, sentence reordering is required.
Honorable Reviewer, thank you for the comment. In the new version of the paper, the conclusions were rewritten.
9.- Check and complete the references.
Honorable Reviewer, thank you for the comment. We understand that we had some missing information and checked all the references. Note that the MDPI template only allows required information to be printed out.
10.- Finally, pay attention to punctuation.
Honorable Reviewer, thank you for the comment. We have made a thorough revision of the English grammar in the new version of the paper including punctuation.
